# Modelling the Influence of Soil Properties on Crop Yields Using a Non-Linear NFIR Model and Laboratory Data

Rebecca L. Whetton [1], Yifan Zhao [2] , Said Nawar [3] and Abdul M. Mouazen [4],*

1    Biosystems Engineering, University College Dublin, D04 V1W8 Dublin, Ireland; Rebecca.whetton@ucd.ie
2    Through-Life Engineering Services Institute, Cranfield University, Bedford MK43 0AL, UK; yifan.zhao@cranfield.ac.uk
3    Soil and Water Department, Faculty of Agriculture, Suez Canal University, Ismailia 41522, Egypt; said.nawar@agr.suez.edu.eg
4    Department of Environment, Faculty of Bioscience Engineering, Ghent University, Coupure Links 653, 9000 Ghent, Belgium
*    Correspondence: Abdul.Mouazen@UGent.be

**Abstract:** This paper introduces a new non-linear correlation analysis method based on a non-linear finite impulse response (NFIR) model to study and quantify the effects of ten soil properties on crop yield. Two versions of the NFIR model were implemented: NFIR-LN, accounting for both the linear and non-linear variability in the system, and NFIR-L, accounting for linear variability only. The performance of the NFIR models was compared with a non-linear random forest (RF) model, to predict oilseed rape (2013) and wheat (2014) yields in one field at Premslin, Germany. The ten soil properties were used as system inputs, whereas crop yield was the system output. Results demonstrated that the individual and total contribution of the soil properties on crop yield varied throughout the different cropping seasons, weather conditions, and crops. Both the NFIR-LN and RF models outperformed the NFIR-L model and explained up to 55.62% and 50.66% of the yield variation for years 2013 and 2014, respectively. The NFIR-LN and RF models performed equally during yield prediction, although the NFIR-LN model provided more consistent results through the two cropping seasons. Higher phosphorus and potassium contributions to the yield were calculated with the NFIR-LN model, suggesting this method outperforms the RF model.

**Keywords:** yield prediction; yield limiting factors; soil fertility; random forest; system identification





## 1. Introduction

Quantification of the soil-related yield limiting factors is essential for site-specific management of farm resources and crop yield forecasts. However, the estimation of these limiting factors is not a straightforward process, as many affecting parameters exist in a very complex system, consisting of the soil and its associated micro- and macro-variability, microclimate, topography, land use, and others. Whilst the relationship between nutrients in the soil and yield is widely understood, there is little work quantifying this relationship using the high sampling resolution data that can be obtained today through the use of advanced remote and proximal sensing technologies. Modelling approaches to predict crop yield have been introduced [1], which included statistical, process-based numerical, machine learning, and parametric modelling approaches. Although statistical models are more suitable for large spatio-temporal scales, they can hardly extrapolate beyond historical extremes [2]. Process-based models [3,4] emulate the main processes of crop growth and development. These models are typically developed and tested using experimental trials, and thus offer the distinct advantage of leveraging decades of research on crop physiology and reproduction, agronomy, and soil science, among other disciplines. These models also require extensive input data on the cultivar, management, and soil conditions that are unavailable in many parts of the world. Apart from overfitting issues, machine learning

modelling techniques were successfully implemented recently to predict wheat yield using proximal sensors [5].

Multiple interacting chemical, biological, and physical factors affect soil fertility and crop yield. In a recent review, the topography of a field was shown to have a significant role in wheat yield variation, due to its influences on soil moisture content and soil properties [6]. Whilst Peralta [7] compared different soil properties to understand field variations in nitrogen (N) demand, they found that the properties that were important for explaining N demand in one field were not important in another field. This highlights that whilst certain factors influencing variation may be obvious, and provide reasonable guidance to field variation, a more detailed understanding may be necessary for accurate precision management.

Parametric models have attracted more and more interest recently due to their limited reliance on field calibration, requiring fewer samples, and their transparent assessment of model performance. Very few studies on the use of parametric modelling to predict yield can be found in the literature. The number of studies on the prediction of crop yield increase when regression analysis is included. Palm [8] produced the simplest form of parametric modelling to estimate crop yield, in the form of a regression of yield against rainfall. In areas where water is a limiting parameter, about 46% of the yield variability was attributed to rainfall. A simple parametric simulation of maize yields using the standard Food and Agriculture Organisation (FAO) methodology was produced by Gommes [9], where 73.75% of the yield variability between separate years was attributed to evapotranspiration.

The non-linear auto-regressive moving average model with eXogenous inputs (NARMAX) is a parametric modelling method introduced by Billings [10]. NARMAX is one of the most popular classes of non-linear system identification methods for a complex system, where the inner structure of the underlying system is unknown but only input and output observational data are available. Compared to machine learning methods, one of the advantages of the NARMAX model is transparency, which means that it can be written down, can be related to known and existing models in the literature, and can also be analysed in frequency and other domains. This characteristic is attractive for studying the brain, climatic change, or an agriculture system. Agriculture is a typical input–output system with an unknown inner structure because it not only allows further frequency analysis or statistical analysis based on the identified model but is also easily understood and interpreted. A non-linear finite impulse response (NFIR) model is a special case of NARMAX that has been recently introduced. Although it has been successfully applied in brain signal analysis [11,12], climate change [13,14], and non-destructive test [15], the model can understand hidden non-linear information, and its application in agriculture is novel.

This paper aims to (1) explore the potential of the newly proposed NFIR models to quantify the soil-related yield limiting factors and their ability to understand the dependence among crop yield and soil properties; and (2) compare the efficiency of the NFIR models with that of random forest (RF) models when identifying and quantifying the most influencing soil properties that limit yield. All methods were tested in one arable field in Germany throughout two cropping seasons, 2013 and 2014, in which oilseed rape and wheat were grown, respectively.

## 2. Materials and Methods

### 2.1. Study Site and Data Collection

The study site was an arable field located in Premslin near Rostock in Germany (Figure 1), with 11°46′ E latitude and 53°6′ N longitude according to the Universal Transverse Mercator (UTM) system. The field is about 33 ha in area, with an average annual rainfall of 591 mm. The average monthly temperature over 10 °C from May to September was of 11.6 °C, 15.3 °C, 17.1 °C, 17.1 °C, and 14.2 °C, respectively. The fields soil was classified as a homogenous Dystric Cambisol (FAO code Bd67-2b) with humic loam texture

on a sandstone rock. The soil was sampled at a depth of 0.2 m, where the clay content was found to vary between 11 and 13%.

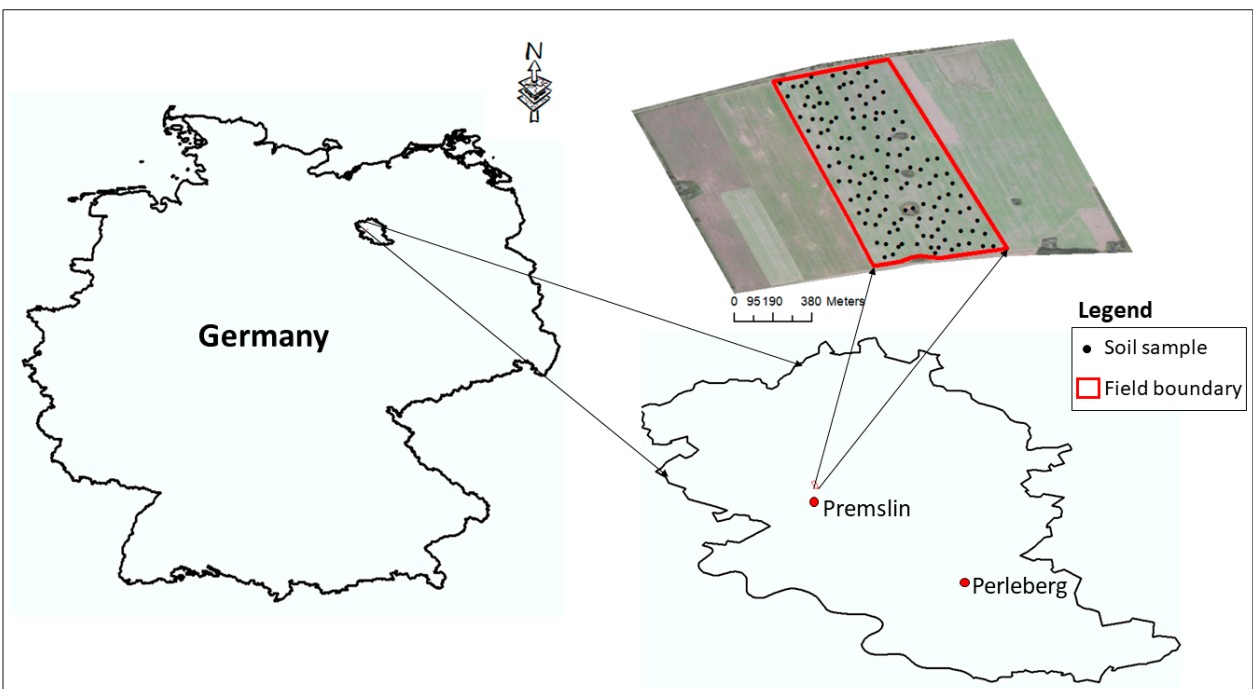

**Figure 1.** Field location in Premslin near Rostock in Germany, where soil samples and yield data were recorded in 2013 and 2014.

A total of 140 soil samples were collected after the harvest of the previous crop in 2013. Composite soil samples of about 700 g were collected over a 1.5 m distance at about 0.15 m depth. An equal gap between neighbouring sampling lines of 12 m was selected. Oilseed rape and wheat were cultivated during the experiment in the 2013 and 2014 cropping seasons, respectively. A differential global positioning system (DGPS) (EZ-Guide 250, Trimble, Sunnyvale, CA, USA) with sub-metre accuracy was used to record the position of each soil sample. The collected soil samples were placed into tightly sealed plastic bags to hold field moisture and stored in a refrigerator at 4 °C, until laboratory analyses. Yield data were collected in 2013 and 2014, at different spatial resolutions to that of the soil samples using the on-board yield sensor of the farmer's combine harvester (New Holland, CX8070 model).

*2.2. Laboratory Analysis and Development of Calibration Models of the Soil Properties*

Each sample was air dried at a temperature not greater than 30 °C, ground, and sieved with a 2 mm sieve. Samples were then subjected to chemical analyses to determine the selected soil parameters that were likely to be limiting the yield of the oilseed rape and wheat. These parameters were pH ($H_2O$), gravimetric soil moisture content (GMC), cation exchange capacity (CEC), organic carbon (OC), total nitrogen (TN), and the available cations sodium ($Na^+$), potassium ($K^+$), calcium ($Ca^{2+}$), magnesium ($Mg^{2+}$), and available phosphorous (P). pH was measured potentiometrically in a suspension with a soil-to-water ratio of 1:2.5 under a controlled temperature environment (MAFF/ADAS, 1986). GMC was determined gravimetrically by oven drying of the soil samples at 105 °C for 24 h. Whilst the moisture content in the soil will vary over short periods of time, the spatial distribution pattern can be assumed to remain more stable. To determine CEC, the soil was saturated with sodium acetate (pH 7.0), and the excess acetate was removed by washing with water and ethanol. The sodium ions absorbed onto the cation exchange sites of the soil were displaced with 1.0 M ammonium acetate, and their concentration was determined using a Flame Photometer [16]. OC was determined according to the British Standard BS 7755

Section 3.8:1995 using a combustion method, which is identical to ISO 10694:1995. TN was determined by the Dumas method, where soil samples are heated to 900 °C in the presence of oxygen gas, as described by British Standard BS EN 13654-2:2001. Exchangeable $K^+$, $Na^+$, $Ca^{2+}$, and $Mg^{2+}$ were extracted with 1.0 M of ammonium nitrate (MAFF/ADAS, 1986), and their concentrations in the extract were determined by Agilent 240 FS AA atomic absorption spectrophotometry (Agilent Technologies, Santa Clara, CA, USA). Available P was determined by extraction from the soil at 20 °C by shaking with 0.5 M sodium bicarbonate solution at pH 8.5 and the concentration was determined by the ascorbic acid method [16].

*2.3. Data Processing*

Yield and ten soil data layers were first fitted to semi-variograms using Vesper software developed by the Australian Centre of Precision Agriculture (Sydney, Australia) to typify the spatial variation. After satisfying semi-variogram selection, the semi-variogram parameters (Table 1) were transferred into ArcGIS (Esri, Redlands, CA, USA) software to perform ordinary kriging to predict the values of the unsampled positions. The krigged data layers were then converted into a common 5 $m^2$ raster grid in ArcGIS (Esri, USA) to assist data fusion [17]. The resulting 5 $m^2$ raster squares of the layers were converted into a common grid of points by extracting the value at the midpoint of each raster square. These steps ensured that all layers consisted of common sets of 5 $m^2$ grid points, which is essential for running different spatial analyses. This method allowed data from a diverse range of soil and yield surveys, measured at different resolutions, to be merged [18]. However, it is worth mentioning that transferring information of a 5 $m^2$ raster to a point would introduce unavoidable error to the spatial distribution of data. The different soil and crop data were subjected to the linear and non-linear NFIR and RF modelling methods, detailed in the following sections.

**Table 1.** Semi-variogram model parameters of the yield data. The best fit was achieved with spherical models.

| Variable | $C_0$ | C | r (m) | ($C_0$/C) (%) | SSE |
|---|---|---|---|---|---|
| Yield 2013 | 0.00029 | 0.00045 | 19.49 | 64.7% | 0.0000 |
| Yield 2014 | 0.00390 | 0.00900 | 40.98 | 43.1% | 0.0001 |

$C_0$ is nugget variance; C is sill; r is the range; and SSE is the sum of squared error.

*2.4. Random Forest Method*

Random forest is a non-linear classification and regression algorithm developed first by Breiman [19], which can be described as follows:

Suppose we have a calibration set $C = \{C_1, \ldots, C_n\}$ with $C_i \equiv (x_i, y_i)$ and an independent test case $C_0$ with predictor $x_0$, the following steps can be carried out:

(1) Sample the calibration set $C$ with replacement to generate bootstrap resamples $B_1, \ldots, B_M$
(2) For each resample $B_m$, $m = 1, \ldots, M$, grow a regression tree $T_m$.
(3) For predicting the test case $C_0$ with covariate $x_0$, the predicted value by the whole RF is obtained by combining the results given by individual trees. Let $\hat{f}^*_m(x_0)$ denote the prediction of $C_0$ by $m^{th}$ tree, the RF prediction for regression problems can then be written as

$$\frac{1}{M} \sum_{m=1}^{M} \hat{f}^*_m(x_0) \tag{1}$$

(4) Parallel to the calibration step, RF performs an internal cross-validation by dividing the calibration set into in-bag (2/3 of the data) and the reaming is assigned to out-of-bag (OOB) sets. The number of variables per level (mtry = 1), size of the nodes (nodesize = 20), and the number of trees (ntree = 500) parameters are optimised by minimising the aggregate error rate of the OOB set (RMSE_OOB) [20,21]. The

accuracy of a random forest's prediction can be estimated from these OOB data, by using the following equation:

$$\text{OOB} - \text{MSE} = \frac{1}{n} \sum_{i=1}^{n} \left( y_{i-} \overline{\hat{y}_{iooB}} \right)^2 \tag{2}$$

where $\overline{\hat{y}_{iooB}}$ is the average prediction for the *i*th observation from all trees, for which this observation has been OOB [22].

Breiman [18] suggested a reduction in MSE (known as variable importance scores) when permuting a variable $X_j$, called "MSE reduction, and decrease in classification accuracy after permuting $X_j$ over all trees". Permutation-based MSE reduction has been adopted as the state-of-the-art approach for measuring the importance of a variable by various authors [23–25]. According to Grömping [20], this is determined as follows: for tree *t*, the OOB mean squared error is calculated as the average of the squared deviations of the OOB responses from their respective predictions:

$$\text{OOB} - \text{MSE}_t = \frac{1}{n\text{OOB}, t} \sum_{\substack{i=1: \\ i \in OOB_t}}^{n} (y_{i-} \hat{y}_{i,t})^2 \tag{3}$$

where the $\hat{y}$ indicates predictions, $\text{OOB}_t$= {*i*: observation *i* is OOB for tree *t*}; that is, the summation is over OOB observations only, and *n*OOB,*t* is the number of OOB observations in tree *t*. If regressor *Xj* does not have a predictive value for the response, it should not make a difference when *Xj* of the OOB data are randomly permuted before calculating the predictions. The OOBMSEt (*Xj* permuted) can be calculated using Equation (4):

$$\text{OOBMSE}t \ (Xj \text{ permuted}) = \frac{1}{n\text{OOB}, t} \sum_{\substack{i=1: \\ i \in OOB_t}}^{n} (y_{i-} \hat{y}_{i,t}(Xj \text{ permuted}))^2 \tag{4}$$

Then, the difference, $\text{OOBMSE}_t$ ($X_j$ permuted) $-$ $\text{OOBMSE}_t$, is calculated for each variable $X_j$ in each tree *t*, based on one permutation of the variable's out-of-bag data for the tree. For the complete forest, the MSE reduction is the average overall *ntree* trees of these differences [22]. The calculated MSE values should be between 0% and 100%. All RF models were performed within the R program using the software package randomForest Version 4.6–12 [26], based on Breiman and Cutler's Fortran code [20]. To assess the results, the MSE reduction was calculated to indicate the contribution of each variable (soil property) on crop yield in 2013 and 2014. The larger the MSE is, the higher dependence is between this term and the output.

### 2.5. Parametric Modelling

In this study, a non-linear finite impulse response (NFIR) model is proposed to understand and quantify the correlation between soil properties and crop yield. This model has been commonly used to represent a multi-inputs and single-output system, and can be expressed as

$$y = f(u_1, u_2, \ldots, u_r) + \varepsilon \tag{5}$$

where *r* is the number of the system inputs; *f* is some unknown linear or non-linear mapping, which links the system output *y* to the system inputs $u_1, u_2, \ldots, u_r$; and $\varepsilon$ denotes the model residual.

A commonly employed model type to specify the function in Equation (5) is a polynomial function [27,28], which can be expressed as follows:

$$y = \theta_0 + \sum_{m=1}^{N} \theta_m \varnothing_m + \varepsilon \tag{6}$$

where $\varnothing_m$ is the $m^{th}$ model term generated from all input vectors; $\theta_m$ is the corresponding unknown parameters; and $N$ is the total number of potential model terms. Note that $\varnothing_m$ is, in general, non-linear. Considering a system with two inputs, $u_1$ and $u_2$, a second-order polynomial function can be written as

$$y = \theta_0 + \theta_1 u_1 + \theta_2 u_2 + \theta_3 u_1^2 + \theta_4 + \theta_5 u_1 u_2 + \varepsilon \tag{7}$$

Comparing with existing linear parametric methods, the proposed method accommodates the non-linear relationship between the inputs and the output by introducing terms $\{u_1^2, u_2^2, u_1 u_2\}$. Note that the candidate term $\varnothing_m$ in Equation (6) can be any linear or non-linear relationship among inputs. This paper will compare the performance of a model accounting only for linear terms, designated as NFIR-L, with a model accounting for both linear and non-linear terms, named NFIR-LN throughout this paper.

The next step is to estimate the parameters $\theta_m (m = 0, 1, \ldots, 5)$ based on the observations $\{y, u_1, u_2\}$. The procedure begins by determining the structure, or the important model terms, using the orthogonal least squares (OLS) estimation procedures. It determines which dynamics and non-linear terms should be included in the model by computing the contribution that each potential model term makes to the variation of the system output. The model is to be built up term by term in a manner that exposes the significance of each new term that is added. Once the structure of the model has been determined, the unknown parameters can be estimated, and the procedure of model validation can ensure the model is adequate. In this paper, a routine called adaptive-forward-orthogonal least squares (AFOLS) was employed, not only to determine the model structure but also to estimate the unknown parameters. A more detailed explanation of this method can be found in Zhao et al. [12].

Finally, the performance of the NFIR models in the prediction of yield was evaluated. This was done by considering the value of error reduction ratio (ERR) for each selected term calculated from AFOLS, which measures the percentage this term contributes to the system output [29]. Values of ERR always range from 0% to 100%. The larger the ERR is, the higher dependence is between this term and the output. It is, therefore, a very important index to indicate the importance of each term to the output.

To calculate the contribution of each input variable to the output, the sum of the ERR values of all selected terms, denoted as $SERR$, is calculated by

$$SERR = \sum_{i=1}^{N} [err]_i \tag{8}$$

to describe the percentage explained by the identified model to the system output, where $N$ denotes the number of the selected terms. If the considered inputs can fully explain the variation of system output, the value of $SERR$ is equal to 100%. It is an indicator of model performance and uncertainty. The contribution of the $i^{th}$ input variable to the variation of the system output, denoted as $ERRC_i$, is defined as the sum of the ERR values of the terms that include this input variable. Because some selected terms may involve more than one input variable due to non-linearity, the sum of $ERRC_i$ for all input variables can be greater than $SERR$. To overcome this problem, the value of $ERRC_i$ is written as

$$ERRC_i = \frac{\sum_{j=1}^{N} \left( [err]_j \big| u_i \in \varnothing_j \right)}{\sum_{p=1}^{r} \sum_{j=1}^{N} \left( [err]_j \big| u_p \in \varnothing_j \right)} \times SERR \tag{9}$$

The value of $ERRC_i$ always should be between 0% and 100%.

## 3. Results and Discussion

In this work, the ten laboratory-measured soil properties (i.e., TN, OC, $Ca^{2+}$, CEC, $Na^+$, $Mg^{2+}$, $K^+$, pH, P, and GMC) were normalised and used as inputs for the NFIR and RF models' establishment, whereas the model output was crop yield. The aim was to investigate the contribution of each soil property on crop yield through two different cropping seasons. Whilst the soil properties were collected in 2013 and yield was collected in 2013 and 2014, it is assumed that the soil properties may have decreased or increased in time.

### 3.1. Soil and Yield Data Analysis

Results of the descriptive statistical analyses of the soil properties are shown in Table 2. The experimental soil ranges from strongly acidic to slightly alkaline (pH = 5.4 to 7.7), with a mean neutral pH value of 6.63. The pH and OM conditions have favourable effects on oilseed rape and wheat growth [30], with a potential pH effect on the P and $Mg^{2+}$ availability for smaller pH ranges than 6.5 [31]. Available $K^+$, $Mg^{2+}$, $Na^+$, and $Ca^{2+}$ are considered low for both crops, with mean values of 0.5 mg/L, 0.29 mg/L, 0.06 mg/L, and 2.86 mg/L, respectively. Available P, CEC, and TN are of low concentrations for oilseed rape and wheat growth [30]. These low concentrations indicate these soil properties to be limiting crop growth and yield. P was consistently one of the largest contributors to yield variation, which could be due to it being in low concentrations for the crops but variable through the field from 9.54 mg/L to 1.34 mg/L.

**Table 2.** Statistics of the measured soil properties of the 140 soil samples used as input data into the three models.

|  | TN | pH | P | OC | GMC | $K^+$ | $Mg^{2+}$ | $Na^+$ | $Ca^{2+}$ | CEC |
|---|---|---|---|---|---|---|---|---|---|---|
| Max | 1.40 | 7.70 | 9.54 | 12.90 | 125.00 | 1.33 | 0.63 | 0.13 | 5.17 | 11.7 |
| Min | 0.50 | 5.40 | 1.34 | 5.90 | 61.50 | 0.14 | 0.13 | 0.02 | 1.47 | 6.30 |
| Mean | 0.80 | 6.63 | 2.69 | 8.30 | 81.90 | 0.50 | 0.29 | 0.06 | 2.86 | 8.45 |
| Std | 0.14 | 0.46 | 0.93 | 1.50 | 11.20 | 0.23 | 0.08 | 0.01 | 0.66 | 1.04 |

OC is organic carbon in %; $K^+$ is exchangeable potassium in mg/L; P is extractable phosphorous in mg/L; GMC is gravimetric soil moisture content in g/kg; TN is total nitrogen in %; CEC is cation exchange capacity in meq/100 g; $Ca^{2+}$ is calcium in mg/L; $Na^+$ is sodium in mg/L; $Mg^{2+}$ is magnesium in mg/L; and pH the log measurement of acidity.

The semi-variogram analysis of the 2013 and 2014 yields shows the best fit of the yield data to be obtained with spherical models, with negligible sums of squares error (SSE) values (Table 1). The degree of spatial dependency for a studied variable can be quantified using the ratio of nugget variance to sill variance [32,33]. Cambardella et al. [32] defined three categories of spatial dependency, these being high, moderate, and weak with ratios of less than 25%, between 25% and 75%, and greater than 75%, respectively. In this study, variations in the ratio of nugget variance to sill variance of the 2013 and 2014 yields are of moderate to high spatial dependence, with ratios of 64.7% and 43.1%, respectively. Ranges of spatial dependence vary from 19.49 m (2013 yield) to 40.98 m (2014 yield), which are wider than the sampling interval of 12 m and confirming the effectiveness of the geostatistical analysis adopted in this study [33]. The yield data in 2014 demonstrated a smaller spatial variability than that in 2013, attributed to different agricultural inputs (fertilisers, pesticides, and seeding rate) applied to different crops.

### 3.2. Linear Correlation

Examining Pearson's correlation coefficient (*r*) values between pairs of soil properties reveals few linear correlations, with the strongest correlation unsurprisingly recorded between OC and TN (*r* = 0.945), which is in line with other reports [34,35]. Other reasonable correlations can be observed between $Ca^{2+}$ and pH (r = 0.716), although those calculated between OC and GMC, TN and $Ca^{2+}$, and TN and GMC are of smaller *r* values, ranging between 0.507 and 0.559. Another interesting but negative linear correlation (*r* = −0.656) is

calculated between pH and CEC, explaining that pH decreases with increasing CEC, which is true as CEC represents the soil's ability to hold positively charged ions, e.g., exchangeable cations [36].

The *r* values indicate no linear correlation could be observed between the ten soil properties and yield in 2013 and 2014 (Table 3). The highest correlation of 0.239 was calculated between OC and the yield of 2013, which is a small value to allow a conclusion to be made on the contribution of soil properties to crop yield. This indicates the system complexity and non-linearity that cannot be explored or quantified by simple linear relationships.

**Table 3.** Pearson correlations (*r*) between the soil properties and yields of 2013 and 2014.

| | $Ca^{2+}$ | CEC | $K^+$ | GMC | $Mg^{2+}$ | $Na^+$ | OC | P | pH | TN | Yield13 | Yield14 |
|---|---|---|---|---|---|---|---|---|---|---|---|---|
| $Ca^{2+}$ | 1.000 | | | | | | | | | | | |
| CEC | −0.018 | 1.000 | | | | | | | | | | |
| $K^+$ | −0.096 | 0.187 | 1.000 | | | | | | | | | |
| GMC | 0.442 | 0.101 | 0.192 | 1.000 | | | | | | | | |
| $Mg^{2+}$ | −0.040 | 0.197 | 0.111 | 0.168 | 1.000 | | | | | | | |
| $Na^+$ | 0.103 | −0.052 | 0.350 | 0.235 | −0.023 | 1.000 | | | | | | |
| OC | **0.507** | 0.150 | 0.004 | **0.510** | 0.189 | −0.033 | 1.000 | | | | | |
| P | −0.238 | 0.180 | 0.074 | −0.360 | −0.219 | 0.004 | −0.131 | 1.000 | | | | |
| pH | **0.716** | **−0.656** | −0.076 | 0.285 | −0.124 | 0.140 | 0.297 | −0.300 | 1.000 | | | |
| TN | **0.564** | 0.216 | 0.079 | **0.559** | 0.217 | 0.006 | **0.945** | −0.130 | 0.305 | 1.000 | | |
| Yield13 | −0.058 | −0.010 | −0.037 | −0.030 | 0.059 | −0.002 | 0.239 | 0.192 | −0.043 | 0.160 | 1.000 | |
| Yield14 | 0.092 | −0.092 | −0.071 | 0.054 | −0.011 | 0.109 | −0.043 | 0.141 | 0.127 | −0.084 | 0.190 | 1.000 |

OC is organic carbon in g/kg; $K^+$ is exchangeable potassium in mg/L; P is extractable phosphorous in mg/L; GMC is gravimetric moisture content in g/kg; TN is total nitrogen in g/kg; CEC is cation exchange capacity in meq/100 g; $Ca^{2+}$ is calcium in mg/L; $Na^+$ is sodium in mg/L; $Mg^{2+}$ is magnesium in mg/L; and pH the log measurement of acidity.

### 3.3. Random Forest Model

To analyse the contribution of the soil variables to the yield prediction, we fit two separate RF models for the dataset of 2013 and 2014. Since the RF algorithm automatically considered the interactions among the explanatory variables, the two separate RF models provided quite similar prediction results for yield, with $R^2$ values of 0.83 and 0.81 for 2013 and 2014, respectively, and both being significant at $p < 0.01$. Permuting the predictor's values over the dataset showed a negative influence on prediction. The MSE with the original dataset were compared with the "permuted" dataset results and the final results are shown in Table 4.

The results show that, according to the MSE values, the soil properties' contribution to the yield are 55.62% and 45.81% for models of the years 2013 and 2014, respectively. A total of 44.38% and 54.19% variation in the yield is left unexplained by both models. This result may be explained by the fact that there are some factors that could not be measured or included in the models, such as weather conditions [37,38] and crop disease [39]. The OC, TN, and $Ca^{2+}$ are the top three highest contributors to oilseed rape yield in 2013, whereas TN, OC, and P are the top three highest contributors to wheat yield in 2014. The OC has the highest variable importance score (12.84%) in 2013; meanwhile, it is the second-highest variable importance score (7.43%) in 2014. This result is expected and supported by the fact that OC can play a vital role in increasing crop yield [40], improving soil fertility [41,42], soil structure [43,44], and water retention [45]. Furthermore, soils with low OC contents have a low crop yield and low use efficiency of added nutrients [40]. Similarly, TN is the highest contributor to yield prediction after OC in 2014, with an MSE value of 11.44%, whereas TN was the second highest contributor in 2013 (MSE = 11.01%). Agegnehu et al. [40] found that nitrogen supply appeared to be a much greater factor limiting yield, which was related to the soil TN content before planting and uptake rate by plants during the growing season. Interestingly, they found that increases in yield and yield components were more pronounced when organic amendments and N fertiliser were both applied, in comparison to one or the other. Surprisingly, $K^+$ has the lowest variable importance score for both

the 2013 and 2014 models, with contributions of 0.82%, and 0.11%, respectively. There is also a gradual decline in the soil $K^+$ levels with crop removal, as cereal removes less $K^+$ compared to other crops [46]. Both P and $Ca^{2+}$ take the third and fourth places on the list. CEC is ranked the fifth and the sixth in 2013 and 2014 with MSE values of 4.73% and 3.23%, respectively. Contrary to CEC, pH is ranked the sixth (3.91%) and fifth (5.12%) on the list for 2013 and 2014, respectively. The remaining soil properties, e.g., $Na^+$, $Mg^{2+}$, and GMC, collectively contribute to the yield of 7.01% and 7.18% for 2013 and 2014, respectively.

**Table 4.** Calculated variable importance score (MSE) of the yield prediction obtained with the random forest (RF) model, indicating the contribution of each soil property on crop yield in 2013 and 2014.

| Rank | 2013 | | 2014 | |
|---|---|---|---|---|
| | Input | MSE (%) | Input | MSE (%) |
| 1 | OC | 12.84 | TN | 11.44 |
| 2 | TN | 11.01 | OC | 7.43 |
| 3 | $Ca^{2+}$ | 9.43 | P | 6.41 |
| 4 | P | 6.22 | $Ca^{2+}$ | 5.20 |
| 5 | CEC | 4.73 | pH | 5.12 |
| 6 | pH | 3.91 | CEC | 3.23 |
| 7 | $Na^+$ | 3.72 | $Mg^{2+}$ | 2.82 |
| 8 | $Mg^{2+}$ | 1.94 | GMC | 2.33 |
| 9 | GMC | 1.35 | $Na^+$ | 2.03 |
| 10 | $K^+$ | 0.82 | $K^+$ | 0.11 |
| Total | | 55.62 | | 45.81 |

OC is organic carbon in g/kg; $K^+$ is exchangeable potassium in mg/L; P is extractable phosphorous in mg/L; GMC is gravimetric soil moisture content in g/kg; TN is total nitrogen in g/kg; CEC is cation exchange capacity in meq/100 g; $Ca^{2+}$ is calcium in mg/L; $Na^+$ is sodium in mg/L; $Mg^{2+}$ is magnesium in mg/L; and pH the log measurement of acidity.

*3.4. Parametric Models*

Based on Equation (6), the following NFIR-LN model with quadratic terms is established to relate the ten soil input variables with yield:

$$y = \theta_0 + \sum_{i=1}^{10} \theta_i u_i + \sum_{i=1}^{10} \sum_{j=i}^{10} \theta_{ij} u_i u_j + \varepsilon \tag{10}$$

This model includes 66 terms consisting of 11 linear terms $\{\theta_0, \theta_i u_i | i = 1, 2, \ldots, 10\}$ and 55 non-linear terms $\{\theta_{ij} u_i u_j | i = 1, 2, \ldots, 10; j = i, i+1, \ldots, 10\}$. The NFIR-L model can be written as

$$y = \theta_0 + \sum_{i=1}^{10} \theta_i u_i + \varepsilon \tag{11}$$

The contributions of each soil property to crop yield variation are listed in Tables 5 and 6 for the NFIR-LN model and NFIR-L model, respectively. The NFIR-LN model accounts for both linear and non-linear interactions, whilst the NFIR-L model only considers linear interactions. The non-linear aspect of the NFIR-LN model allows for more understanding of the yield variation. This is confirmed by the SERR values for 2013 and 2014 being 52.23% and 50.66% for the NFIR-LN model, and 19.15% and 8.5% for the NFIR-L model, respectively. This is supported by the negligible linear correlations calculated (*r* values; Table 3). The variation in SERR contribution between the years can be attributed to varying weather conditions [47,48], pests [39,49], and, finally, the different crops grown, e.g., oilseed rape in 2013 and wheat in 2014.

**Table 5.** Calculated individual contribution (ERRC) of each soil property and the sum of contribution (SERR) to crop yield in the 2013 and 2014 cropping seasons, obtained with the non-linear finite impulse response (NFIR) model, accounting for both linear and non-linear relationships (NFIR-LN).

| Rank | 2013 | | 2014 | |
|---|---|---|---|---|
| | Input | ERRC (%) | Input | ERRC (%) |
| 1 | P | 12.74 | P | 8.30 |
| 2 | CEC | 12.47 | $Na^+$ | 6.77 |
| 3 | OC | 6.44 | OC | 6.71 |
| 4 | pH | 4.91 | $K^+$ | 5.89 |
| 5 | $Ca^{2+}$ | 3.87 | pH | 5.05 |
| 6 | $Na^+$ | 3.82 | TN | 5.01 |
| 7 | $Mg^{2+}$ | 3.41 | GMC | 5.00 |
| 8 | $K^+$ | 1.75 | $Mg^{2+}$ | 4.06 |
| 9 | GMC | 1.59 | CEC | 2.79 |
| 10 | TN | 1.23 | $Ca^{2+}$ | 1.08 |
| SERR | | 52.23 | | 50.66 |

OC is organic carbon in g/kg; $K^+$ is exchangeable potassium in mg/L; P is extractable phosphorous in mg/L; GMC is gravimetric soil moisture content in g/kg; TN is total nitrogen in g/kg; CEC is cation exchange capacity in meq/100 g; $Ca^{2+}$ is calcium in mg/L; $Na^+$ is sodium in mg/L; $Mg^{2+}$ is magnesium in mg/L; and pH the log measurement of acidity.

**Table 6.** Calculated individual contribution (ERRC) of each soil property and the sum of contribution (SERR) to crop yield in the 2013 and 2014 cropping seasons, obtained with the non-linear finite impulse response (NFIR) model, accounting for a linear relationship only (NFIR-L).

| Rank | 2013 | | 2014 | |
|---|---|---|---|---|
| | Input | ERRC (%) | Input | ERRC (%) |
| 1 | P | 6.18 | P | 2.01 |
| 2 | OC | 5.73 | OC | 1.46 |
| 3 | TN | 3.94 | $Na^+$ | 1.21 |
| 4 | $Ca^{2+}$ | 1.23 | $Ca^{2+}$ | 1.20 |
| 5 | pH | 1.09 | $K^+$ | 1.13 |
| 6 | $Mg^{2+}$ | 0.53 | TN | 0.88 |
| 7 | $K^+$ | 0.17 | CEC | 0.33 |
| 8 | $Na^+$ | 0.15 | GMC | 0.18 |
| 9 | GMC | 0.13 | $Mg^{2+}$ | 0.07 |
| 10 | CEC | 0 | pH | 0.03 |
| SERR | | 19.15 | | 8.50 |

OC is organic carbon in g/kg; $K^+$ is exchangeable potassium in mg/L; P is extractable phosphorous in mg/L; GMC is gravimetric soil moisture content in g/kg; TN is total nitrogen in g/kg; CEC is cation exchange capacity in meq/100 g; $Ca^{2+}$ is calcium in mg/L; $Na^+$ is sodium in mg/L; $Mg^{2+}$ is magnesium in mg/L; and pH the log measurement of acidity.

The output of the NFIR-LN model (ERRC) shows P, CEC, and OC to be the top three contributors to oilseed rape yield variability in 2013, whereas P, $Na^+$, and OC are the top three contributors to wheat in 2014. This seems logical from the soil fertility point of view, as these (apart from $Na^+$) are all key nutrients to crop growth and development [50]. As indicated above, OC can improve soil fertility, soil structure, and water retention, and hence improve soil productivity. $Na^+$ is a minor nutrient that can negatively affect the moisture uptake and inhibit enzyme activities at high levels. The high yield variability attributed to $Na^+$ from the NFIR-LN models is thus unexpected. Though there is an optimal K:Na ratio for plant growth and yield development [51]. High amounts of K have been found to reduce the occurrence of crop disease. High quantities of Na can reduce the availability of K to plants [52] Within the studied field, levels of $K^+$ were more variable through the field than $Na^+$; the interaction between $K^+$ and $Na^+$ may help explain the higher $Na^+$ impact on yield variation, particularly in the NFIR-LN model [50,51] (Wakeel 2013). CEC is often

used as a measurement of soil fertility, but is not nutrient specific, whereas P is the main soil nutrient (with $N^+$ and $K^+$) for crop growth and development.

Surprisingly, TN has a small contribution to yield variation in 2013 (ERRC = 1.23%), which can be attributed to the small variation in nitrogen in this field (Table 2), or the smaller sensitivity of oilseed rape to nitrogen as compared to wheat, as TN demonstrates a higher contribution (ERRC = 5.01%) in 2014′s calculations. Both pH and $Mg^{2+}$ are quite consistent contributors to the 2013 and 2014 yield variability with the calculated ERRC values of 4.915 and 5.05%, respectively. The range of soil pH in this field is between moderately basic to moderately acidic (Table 2). Furthermore, it is interesting to see that GMC has a low contribution to yield variation in 2013 (ERRC = 1.59%), with a higher contribution calculated in 2014 (ERRC = 5.0%), which may be attributed to different crop or weather conditions across the two cropping seasons. The calculated contributions of $K^+$ to the yield variability was the highest contribution in 2014. $Ca^{2+}$ is the lowest contributor in 2014 only.

The NFIR-L model shows different ranking results and different contributions of individual soil properties as compared to the NFIR-LN model. With the NFIR-L model, both P and OC are the top two contributors to yield variability in both the 2013 and 2014 years. Contrary to the NFIR-LN model, TN obtained with the NFIR-L model in 2013 is of more significant influence and records the same ranking in 2014, although the ERRC with the NFIR-L model is smaller (e.g., 0.88) compared to the NFIR-LN model (e.g., 5.01%), which is attributed to the larger SERR of the latter than the former approach. $Ca^{2+}$ has a remarkably similar contribution in both 2013 and 2014, being the fourth most significant parameter. $Na^+$ is a larger contributor to yield variability in 2014 than in 2013, which is in line with the NFIR-LN model predictions. The pH significantly varies in contribution between the two years; this could be due to the availability of nutrients detailed in Table 2, being reasonably neutral in acidity. $Mg^{2+}$ and GMC are both insignificant contributors, which is in line with the NFIR-LN model that accounts for both the linear and non-linear variability. $K^+$ holds a similar position in 2013 and 2014 for both parametric models investigated. CEC most interestingly has no effect in 2013 on the NFIR-L model, opposed to the NFIR-LN model predictions.

*3.5. Parametric Models versus Random Forest Models*

Based on the results discussed above, it seems that OC always retains a high contribution to yield variability through both the years. This is true for all three models investigated in this study. Soil OC arguably is the best single indicator of soil quality and function because of its impact on soil physical, chemical, and biological properties [53–55]. Furthermore, OC is a source of plant nutrients in soils and is vital in maintaining and improving soil structure, promoting water retention and reducing erosion [55]. Therefore, it is not surprising to observe that soil OC is a high contributor to yield variability. P has a direct link with yield and so is expected to be a high contributor to yield variability [56]. P was the highest contributor to yield variability in both NFIR models. However, with the RF model, P consistently contributed around 6% to yield variability, being ranked as the third contributor to yield variability in 2014 and fourth in 2013. Since N is a key indicator for soil quality and plays a vital role in crop production [57], this is an advantage of RF over NFIR. However, RF would have a stronger weight if mineral nitrogen was considered in the current work instead of TN [58].

Although the NFIR-LN and RF models allow the prediction of yield variability to be around 50%, with the input data from ten soil properties, there is a remaining 50% of the contribution that has not been accounted for. Kravchenko and Bullock [59] stated that yield variability is caused by a host of factors in addition to topographical and soil characteristics. Therefore, it is suggested that there are other external factors (such as micro climate conditions, pests, and compaction) that need to be incorporated along with soil properties to allow for a more accurate assessment of the yield limiting factors and for improving the prediction accuracy of yield. P along with N are commonly added as fertilisers, and are the

main soil macro nutrients, affecting the severity of fungal diseases [60–63]. Whilst fungal diseases have not been considered in this study, it would be interesting to involve them in further work, particularly with the yield variation attributed to Na by the NFIR models.

The total contributions of the RF model to crop yield variability are 55.6% and 45.8% in the 2013 and 2014 cropping seasons, respectively. The NFIR-LN model that is more successful than the NFIR-L model in yield variability prediction contributes with similar percentages (52.23% and 50.66% for years 2013 and 2014, respectively) to those of the RF models. However, the total contribution of the NFIR-LN models seems more consistent and stable, as the difference between the 2013 and 2014 prediction is 1.57%, whereas the difference for the RF models is greater at 9.8%.

Consistency in the contribution of the regression models to predict yield was an issue raised by Kravchenko and Bullock [60]. Their findings suggested that the capability of models to predict yield based on input soil properties is heavily variable; whilst successful in explaining a substantial portion of the yield in some years, it is only capable of explaining a small portion of the yield variability in others. Therefore, consistency in the model's predictions is highly valued. Each of the models demonstrated some consistency between the two years, with RF being more consistent in ranking the influencing soil properties on yield variability. The NFIR-LN predicted a much larger contribution of $K^+$ to yield variation than the RF model, particularly in 2014. K is one of the main nutrients affecting crop growth and is correlated to disease pressure [50], making the NFIR-LN model an interesting concept for further work, if disease data could be included. Furthermore, the largest calculated P contribution obtained with the NFIR-LN model as compared to the RF model (Tables 4 and 5), with P having a direct link with crop yield, presents strong evidence of NFIR-LN out-performing RF models. As compared to the RF modelling output, NFIR-LN predicted a much larger contribution of $K^+$ to yield, particularly in 2014, providing another clue that NFIR-LN is a better predictor of crop responses.

## 4. Conclusions

To quantify the soil-related yield limiting factors, a new parametric modelling technique based on a non-linear finite impulse response (NFIR) technique was applied and compared with the random forest technique. The input variables were ten soil properties, namely, total nitrogen (TN), organic carbon (OC), potassium (K), pH, phosphorous (P), gravimetric soil moisture content (GMC), calcium (Ca), CEC, sodium (Na), and magnesium ($Mg^{2+}$), and the model output was crop yield. The analysis was carried out in one field in Germany through two cropping seasons, 2013 (oilseed rape) and 2014 (wheat).

The results of the NFIR-LN model, accounting for both linear and non-linear interactions, explained 52.23% and 50.66% of the yield variation in 2013 and 2014, respectively, which was higher than those obtained with the NFIR-L model (total contributions of 19.15% and 8.5% in 2013 and 2014, respectively), accounting for the linear interaction only. The contribution of the RF model produced the highest contribution of 55.6% in 2013, which dropped down to 45.8% in 2014. In this research, the NFIR-LN model indicated that the P, CEC, and OC are the highest contributors to oilseed rape yield in 2013 and P, $Na^+$, and OC to wheat yield in 2014. TN was surprisingly a small contributor, particularly in 2013. It was observed that P and OC are consistently the highest contributors to yield through all NFIR models. The RF analysis presented OC and TN to be the highest contributors to yield in both studied cropping seasons.

The higher P and $K^+$ contributions to yield calculated with the NFIR-LN model, and consistency in overall contributions through the two years, as compared to the RF models, suggested a more stable and robust performance of the NFIR-LN models as compared to the RF models. For the two studied years, this study could draw an understanding of the factors that influence ~50% of the variability in yield in this field. Therefore, we recommend the new proposed NFIR-LN method to model and quantify the influences of soil-related yield limiting factors. However, further examination is advocated in different crops and soil types. More importantly, soil physical properties (e.g., soil compaction, hydraulic

conductivity, etc.), environmental factors, pest, and land attributes should be considered in the analysis to have a full picture when quantifying the site-specific yield limiting factors; i.e., regarding the whole system. High-resolution data on soil physical and chemical properties would be interesting to investigate to allow for site-specific management of land resources.

**Author Contributions:** The specific division and contributions of work from the authors to the manuscript are as follows; software, and methodology, S.N. and Y.Z.; formal analysis, investigation, visualization, data curation, validation, S.N., Y.Z. and R.L.W.; writing—original draft preparation, R.L.W.; writing—review and editing, resources, conceptualization, supervision, project administration, and funding acquisition, A.M.M. All authors have read and agreed to the published version of the manuscript.

**Funding:** We acknowledge the funding received for Farm FUSE project from the ICT-AGRI under the European Commission's ERA-NET scheme under the 7th Framework Programme, and the UK Department of Environment, Food and Rural Affairs (contract No: IF0208).

**Institutional Review Board Statement:** Not applicable.

**Informed Consent Statement:** Not applicable.

**Data Availability Statement:** Data available on request.

**Conflicts of Interest:** The authors declare no conflict of interest. The funders had no role in the design of the study; in the collection, analyses, or interpretation of data; in the writing of the manuscript, or in the decision to publish the results.

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
