# Peer review of "Modelling the Influence of Soil Properties on Crop Yields Using a Non-Linear NFIR Model and Laboratory Data"

_soilsystems, doi:10.3390/soilsystems5010012_

Round 1
Reviewer 1 Report
Review for soilsystems-1075581" NFIR-model-based nonlinear correlation analysis of soil properties on crop yields based on a limited number of laboratory data "by Whetton et al. The manuscript was well written, discussed and presented. However, there are some minor comments and corrections before the acceptance.
Introduction
This section can be extended; since the paper aims to monitor and evaluate the soil properties' effect on crop yield, it is better to explain the soil variations and their variability in the farms and landscape. Moreover, the yield production directly depends on soil quality distribution. If the expert does not have a general overview of the farms regarding the soil quality variability, maybe s/he could not realize some model's over-under-estimation. So it is recommended to add a paragraph about their variability. The following paper can be helpful.
Spatial Variability of Rainfed Wheat Production Under the Influence of Topography and Soil Properties in Loess‑Derived Soils, Northern Iran. INTERNATIONAL JOURNAL OF PLANT PRODUCTION.
L11: nonlinear or non-linear, Oilseed or Oil seed, please keep constant during the writing.
L44-45: please avoid 1-2 sentences as a paragraph.
Figures 1, 2 it not clear. Low quality. Increase the size and resolution.
L83: suggestion: it is better to merge Figs 1 and 2. It is more appealing if you can present sampling points in the real color image.
Author Response
Authors: would like to thank both the guest editor and the three reviewers for the invaluable comments that have led to considerable improvement in the manuscript. We have now completely revised the manuscript. In the following, we respond to the individual remarks of the reviewers.
Reviewer 1: This section can be extended; since the paper aims to monitor and evaluate the soil properties' effect on crop yield, it is better to explain the soil variations and their variability in the farms and landscape. Moreover, the yield production directly depends on soil quality distribution. If the expert does not have a general overview of the farms regarding the soil quality variability, maybe s/he could not realize some model's over-under-estimation. So it is recommended to add a paragraph about their variability. The following paper can be helpful.
Spatial Variability of Rainfed Wheat Production Under the Influence of Topography and Soil Properties in Loess‑Derived Soils, Northern Iran. INTERNATIONAL JOURNAL OF PLANT PRODUCTION.
Authors: An additional paragraph has been included in the introduction (lines )
Reviewer 1: L11: nonlinear or non-linear, Oilseed or Oil seed, please keep constant during the writing.
Authors: This has been considered through the manuscript.
Reviewer 1: L44-45: please avoid 1-2 sentences as a paragraph.
Authors: This was part of the previous page, but I will ensure it doesn’t occur in the manuscript.
Reviewer 1: Figures 1, 2 it not clear. Low quality. Increase the size and resolution.
Authors: Thank you. Fig. 1 has been improved in the revised version.
Reviewer 1: L83: suggestion: it is better to merge Figs 1 and 2. It is more appealing if you can present sampling points in the real color image.
Authors: Thank you. Fig. 1 has been improved to present the soil samples as requested.
Reviewer 2 Report
1) New model NFIR is proposed to quantify soil related yield limiting 66 factors and understand the dependence among crop yield and soil properties
2) Paper is well written
3) Comparison of NFIR with only random forest is done. NFIR model can be compared with other models
4) Experiments were conducted on only 2 datasets (oil seed rape and wheat). Experiments on multiple datasets could have been conducted.
5) Multiple performance metrics could have been used to compare the models.
6) Justification of why the proposed method (NFIR) performs well is required
Author Response
Authors: would like to thank both the guest editor and the three reviewers for the invaluable comments that have led to considerable improvement in the manuscript. We have now completely revised the manuscript. In the following, we respond to the individual remarks of the reviewers.
Reviewer 2: 1) New model NFIR is proposed to quantify soil related yield limiting 66 factors and understand the dependence among crop yield and soil properties
Reviewer 2: 2) Paper is well written
Authors: Thank you for your constructive opinion.
Reviewer 2: 3) Comparison of NFIR with only random forest is done. NFIR model can be compared with other models
Authors: The main objective of the current research was to compare the performance of two versions of NFIR model with a power full regression method (random forest, RF) for predicting the yield of oilseed rape and wheat. NFIR models, namely, NFIR-LN, account for both the linear and non-linear variability in the system, and NFIR-L, account for linear variability only. RF method has advantages over other multivariate analysis. RF can model both linear and nonlinear relationships outperforming other multivariate methods like PLSR, gradient boosted machines (GBM), and artificial neural networks (ANNs) (Nawar et al., 2019; Douglas et al., 2019; Nawar and Mouazen, 2017). RF doesn’t require a specific data pre-treatment (e.g., scaling or transformation) and runs very fast (Nawar et al., 2019).
Nawar, S., Mouazen, A.M. 2017. Comparison between random forests, artificial neural networks and gradient boosted machines methods of on-line vis-NIR spectroscopy measurements of soil total nitrogen and total carbon. Sensors 2017, 17, 2428; doi:10.3390/s17102428.
Douglas, R.K., Nawar, S., Coulon, F., Mouazen, A.M. 2019. Rapid detection of alkanes and polycyclic aromatic hydrocarbons in oil-contaminated soils using visible near-infrared spectroscopy. European Journal of Soil Science, 70, 140-150.
Nawar, S., Ynse, D., Delbecque, N., De Grave, J., De Smedt, P., Finke, P., Verdoodt, A., Van Meirvenne, M., Mouazen, A.M. 2019. Can spectral analyses improve measurement of key soil fertility parameters with X-ray fluorescence spectrometry? Geoderma, 350 (2019) 29–39.
Reviewer 2: 4) Experiments were conducted on only 2 datasets (oil seed rape and wheat). Experiments on multiple datasets could have been conducted.
Authors: Due to the limited budget of the project, two crops were selected in the current research. However, future work should focus on testing the obtained results with multiple datasets.
Reviewer 2: 5) Multiple performance metrics could have been used to compare the models.
Authors: To compare between the models, the simple and recommended reduction in RMSE of prediction was used to compare the performance of NFIR-LN and NFIR-L models with the performance of RF machine learning models. However, this comment will be considered in future research.
Reviewer 2: 6) Justification of why the proposed method (NFIR) performs well is required
Authors: The paper has been edited to better defend the conclusion, and include a bit more
Reviewer 3 Report
Manuscript ID: soilsystems-1075581-peer-review-v1
Type of manuscript: Article
Title: NFIR-model-based nonlinear correlation analysis of soil properties on crop yields based on a limited number of laboratory data
Major Comments:
In my opinion, this article has two global problems.
First, it is associated with this type of work, when a deliberately unacceptable comparison option or methods are taken and on their basis the results already understood by others are proved. In this case, the function, response (yield), according to Liebig's law of limiting factors, is a priori related to the variables by a nonlinear relationship (linear correlation can be obtained only as a special case when empirical data are scarce). Why use Pearson correlation (Table 3), which is for qualitative or pseudo-quantitative features and has rough estimates? Why, instead of the correlation coefficient for linear regression, not calculate the correlation ratio for nonlinear relationships? Why don't the authors go straight to the innovations in multivariate nonlinear analysis, improving methods, especially models of the genetic kind (and not polynomials of the n-th degree up to infinity, which can approximate anything, but work only according to the amount of data entered?
Second, the random selection of soil indicators to relate to yield is surprising. There are more conservative indicators, like OC, TN, CEC, MC, and there are labile ones like labile humus, pH, mobile phosphorus, which, as appropriate, is used. In the same time, why were not readily hydrolysable nitrogen and exchangeable potassium determined? The harvest for the year looks strange, but there is no NPK content (agronomic axiom). The authors have combined everything "in a heap". I was particularly impressed with the use of MC, for some reason named moisture content. MC content is an estimate of hygroscopic moisture as described in the methods and characterizes the basic matrix of the soil system (mineralogy and particle size distribution).
And finally, third, if the title of the article says “based on a limited number of laboratory data”, then what is so little? 140 soil samples on 33 ha allowed in Fig. 2 to represent visualization using geostatistics. I see a contradiction here.
Specific Comments:
A4 Abdul M.
L38 [3.4]
L69-70. All methods were tested in one arable field in Germany, throughout two cropping seasons in 2013 and 2014 with oil seed rape 69 and wheat grown, respectively.
This sentence is pertinent on L 94-95.
L75. It is better to give the sum of temperatures over 10 degrees during the growing season.
L77. A more accurate particle size distribution is needed, because on 33 hectares, judging by the MC variation, it was not the same.
L88. Here it is necessary to indicate agricultural crops in 2013 and 2014.
L92. Not early presented Fig. 2 with a description of methods that will become clear only with L 111?
L97. It should be indicated that pH (H2O), although your variation was from acidic to alkaline, and therefore pH (KCl)
L 98. Cations should be given with an indication of the valency. This is also in Table 1.
L100. Somewhere there is a meaningful interpretation of the choice of MC as a parameter and its role in the soil system? How many particles were <0.002 mm? What can explain such a spread in MS values (Table 1).
L460-61. Name error: Yifan Zhao;
L 465 Error ::
L466-71. One phrase is enough (in quotes)
References:
There are names with capital letters (No. 3, 21 ...)
It's not clear what these abbreviations are? No. 30, 31, 54.
L597-598. As they say in England, "this is a tail from another cat" :)
Author 1, A.B. (University, City, State, Country); Author 597 2, C. (Institute, City, State, Country). Personal communication, 2012.
Author Response
Authors: would like to thank both the guest editor and the three reviewers for the invaluable comments that have led to considerable improvement in the manuscript. We have now completely revised the manuscript. In the following, we respond to the individual remarks of the reviewers.
Reviewer 3: In my opinion, this article has two global problems.
First, it is associated with this type of work, when a deliberately unacceptable comparison option or methods are taken and on their basis the results already understood by others are proved. In this case, the function, response (yield), according to Liebig's law of limiting factors, is a priori related to the variables by a nonlinear relationship (linear correlation can be obtained only as a special case when empirical data are scarce). Why use Pearson correlation (Table 3), which is for qualitative or pseudo-quantitative features and has rough estimates? Why, instead of the correlation coefficient for linear regression, not calculate the correlation ratio for nonlinear relationships? Why don't the authors go straight to the innovations in multivariate nonlinear analysis, improving methods, especially models of the genetic kind (and not polynomials of the n-th degree up to infinity, which can approximate anything, but work only according to the amount of data entered?
Authors: RF can be described as the sum of piecewise linear functions in contrast to the global linear and polynomial regression models. Consequently, RF via the decision tree algorithm, subdividing the input dataset into smaller sets that become more manageable. RF variable importance has been used at each split, to calculate how much this split reduces node impurity (for regression trees, indeed, the difference between RSS before and after the split). This is summed over all splits for that variable, overall trees. In other words, if a predictor (e.g., soil property) is important in the yield prediction model, then assigning other values for that predictor randomly but 'realistically' (i.e.: permuting this predictor's values over the dataset), should have a negative influence on prediction, i.e.: using the same model to predict from data that is the same except for the one variable, should give worse predictions. This has been mentioned in the manuscript (lines 278-280) ‘’Since the RF algorithm automatically considered interactions among the explanatory variables, the two separate RF models provided quite similar prediction results for yield with R2 of 0.83 and 0.81 for 2013 and 2104, respectively. Permuting the predictor's values over the dataset showed a negative influence on prediction.’
Pearson correlation is a well-accepted method to investigate the linear correlation between one soil property and the yield, which was used in this paper as the baseline. Moreover, this method also allows us to analyse the correlation between any two soil properties. The result suggests that there is some redundant information in the inputs, such as OC vs TN. They are not orthogonal. We need a more advanced approach to better understand how the soil properties affect the yield. Furthermore, Pearson correlation is a non-parametric method, which measures the similarity of data pattern directly, while NFIR is a parametric-model, which establishes the relationship between the inputs and output. The sum of the contribution of each soil property in Table 4 and 5 is always no more than 1, which is not the case for the Pearson correlation. Collectively, the authors present three different methods to demonstrate how the proposed method can help better understand this relationship.
Reviewer 3: Second, the random selection of soil indicators to relate to yield is surprising. There are more conservative indicators, like OC, TN, CEC, MC, and there are labile ones like labile humus, pH, mobile phosphorus, which, as appropriate, is used. In the same time, why were not readily hydrolysable nitrogen and exchangeable potassium determined? The harvest for the year looks strange, but there is no NPK content (agronomic axiom). The authors have combined everything "in a heap". I was particularly impressed with the use of MC, for some reason named moisture content. MC content is an estimate of hygroscopic moisture as described in the methods and characterizes the basic matrix of the soil system (mineralogy and particle size distribution).
Authors: A few details of P and K are added into the discussion (lines )
We selected these soil properties because they can be measured with our on-line vis-NIR sensor that provides data with high resolution. The mineral composition or mobile forms of soil nutrients can not be measured with the Vis-NIR tool. we used the Olson P method, in the analysis. We have to say this is the gravimetric moisture content measured by over-drying at 105 deg, for 24 h.
Reviewer 3: And finally, third, if the title of the article says “based on a limited number of laboratory data”, then what is so little? 140 soil samples on 33 ha allowed in Fig. 2 to represent visualization using geostatistics. I see a contradiction here.
Authors: The title has been modified.
Reviewer 3: Specific Comments:
Reviewer 3: L38 [3.4]
Authors: changed.
Reviewer 3: L69-70. All methods were tested in one arable field in Germany, throughout two cropping seasons in 2013 and 2014 with oil seed rape 69 and wheat grown, respectively.
This sentence is pertinent on L 94-95.
Authors: The correction has been made.
Reviewer 3: L75. It is better to give the sum of temperatures over 10 degrees during the growing season.
Authors: Extra detail has been included, however, exact temperatures were not recorded at the field location, here, the greater specification may be misleading.
Reviewer 3: L77. A more accurate particle size distribution is needed, because on 33 hectares, judging by the MC variation, it was not the same.
Authors: Extra details have been added (lines )
Reviewer 3: L88. Here it is necessary to indicate agricultural crops in 2013 and 2014.
Authors: Crop details were moved
Reviewer 3: L92. Not early presented Fig. 2 with a description of methods that will become clear only with L 111?
Authors: Figure 2 has been moved to section 2.3
Reviewer 3: L97. It should be indicated that pH (H2O), although your variation was from acidic to alkaline, and therefore pH (KCl)
Authors: added
Reviewer 3: L 98. Cations should be given with an indication of the valency. This is also in Table 1.
Authors: added
Reviewer 3: L100. Somewhere there is a meaningful interpretation of the choice of MC as a parameter and its role in the soil system? How many particles were <0.002 mm? What can explain such a spread in MS values (Table 1).
Authors: The soil samples were collected every 12 m across the field, which could cover different areas of compaction, additional soil texture information has been given.
Reviewer 3: L460-61. Name error: Yifan Zhao;
Authors: the correction has been made.
Reviewer 3: L 465 Error ::
Authors: corrected
Reviewer 3: L466-71. One phrase is enough (in quotes)
Authors: Shortened to just the statements.
Reviewer 3: References:
There are names with capital letters (No. 3, 21 ...)
Authors: A few titles were capitalised due to journal style, I have changed these for the manuscript
It's not clear what these abbreviations are? No. 30, 31, 54.
Authors: A formatting issue occurred, and I have corrected it
Reviewer 3: L597-598. As they say in England, "this is a tail from another cat" :)
Authors: Thank you. This was a remnant from the paper format.
Round 2
Reviewer 3 Report
Comment for Authors:
Determination of soil moisture content dynamics is an important procedure for regular monitoring and programming of crops, for example, in semi-arid and arid agricultural landscapes for assessing irrigation rates. When the article is based on the data of the harvest of individual years, the reader may involuntarily perceive such a term as "soil moisture content (SMC)" in the traditional way, as the moisture content at a given time (period), and not as a constant. If the authors in their response to the reviewer agreed that the content of the SMC indicator is "the gravimetric moisture content", then it would be correct to indicate this throughout the text of the article and change the abbreviation.
Author Response
Thankyou for your suggestion. We have changed the abbreviation of SMC to GMC, and included additional information on lines 106 and 107.